# GSF-LLM: Graph-Enhanced Spatio-Temporal Fusion-Based Large Language Model for Traffic Prediction

**DOI:** 10.3390/s25216698

**Published:** 2025-11-02

**Authors:** Honggang Wang, Ye Li, Wenzhi Zhao, Haozhe Zhu, Jin Zhang, Xuening Wu

**Affiliations:** 1Urban Mobility Institute, Tongji University, Shanghai 200092, China; wanghonggang@tongji.edu.cn; 2College of Transportation Engineering, Tongji University, Shanghai 200092, China; 2431768@tongji.edu.cn; 3Khoury College of Computer Science, Northeastern University, Boston, MA 02115, USA; zhu.haozh@northeastern.edu; 4School of Computer and Information Engineering, Henan University, Kaifeng 475000, China; zhangjin@henu.edu.cn; 5College of Intelligent Robotics and Advanced Manufacturing, Fudan University, Shanghai 200433, China; xueningwu@gmail.com

**Keywords:** traffic prediction, large language model, spatial-temporal data

## Abstract

Accurate traffic prediction is essential for intelligent transportation systems, urban mobility management, and traffic optimization. However, existing deep learning approaches often struggle to jointly capture complex spatial dependencies and temporal dynamics, and they are prone to overfitting when modeling large-scale traffic networks. To address these challenges, we propose the GSF-LLM (graph-enhanced spatio-temporal fusion-based large language model), a novel framework that integrates large language models (LLMs) with graph-based spatio-temporal learning. GSF-LLM employs a spatio-temporal fusion module to jointly encode spatial and temporal correlations, combined with a partially frozen graph attention (PFGA) mechanism to model topological dependencies while mitigating overfitting. Furthermore, a low-rank adaptation (LoRA) strategy is adopted to fine-tune a subset of LLM parameters, improving training efficiency and generalization. Experiments on multiple real-world traffic datasets demonstrate that GSF-LLM consistently outperforms state-of-the-art baselines, showing strong potential for extension to related tasks such as data imputation, trajectory generation, and anomaly detection.

## 1. Introduction

Traffic flow prediction aims to estimate the future utilization of transportation resources in various urban regions over a given time horizon (e.g., the next hour or several hours) [1]. For instance, it can be used to forecast the demand for taxis or shared bicycles in different districts. Accurate traffic forecasting is essential for optimizing traffic control strategies and public transportation scheduling [2,3,4], and it serves as a foundational component of intelligent transportation systems (ITSs) [5].

Traditional traffic forecasting methods primarily rely on time series models such as the Kalman filter and the autoregressive integrated moving average (ARIMA) model [6]. Although these approaches are effective in capturing temporal dependencies, they are often inadequate for modeling the complex spatio-temporal correlations that characterize traffic data. In recent years, the rise of deep learning has led to the development of various neural network-based models, including convolutional neural networks (CNNs), recurrent neural networks (RNNs), graph convolutional networks (GCNs), and attention-based frameworks, which are designed to extract both spatial and temporal features from traffic observations [7,8,9]. These models typically handle spatial and temporal dependencies through sequential, parallel, or decoupled processing strategies [10,11,12]. However, despite continuous advances in architectural design, the improvement in predictive performance has gradually reached a plateau.

In recent years, large language models (LLMs) have achieved remarkable success across a wide range of domains, including natural language processing and computer vision [13,14]. Compared to conventional neural architectures, LLMs possess superior representation capabilities and can be adapted to new tasks with minimal fine-tuning, thereby eliminating the need for extensive architectural modifications. Motivated by these advantages, an increasing number of studies have investigated the application of LLMs to time series forecasting, covering both short-term and long-term prediction tasks. However, in long-horizon scenarios such as traffic flow forecasting, existing LLM-based methods have yet to achieve satisfactory performance [15,16,17,18]. This limitation is primarily due to two factors. First, the high computational costs associated with LLM training and inference hinder their practical deployment in resource-constrained environments. Second, there exists a substantial domain gap between natural language and structured traffic data, which reduces the transferability of pre-trained models.

To alleviate the computational burden, some studies adopt a frozen pre-trained (FPT) fine-tuning strategy, freezing core LLM components like feed-forward and multi-head attention modules during training. While this reduces overhead and improves generalization in traffic tasks, most existing approaches treat spatial and temporal embeddings as separate, independent features, largely ignoring their complex interactions. To our knowledge, only a few attempts, such as ST-LLM [19] and ST-LLM++ [20], have addressed this interaction by concatenating embeddings followed by pointwise convolution. Although these increase representational capacity, they fail to explicitly model the cross-dependencies among spatial, temporal, and node-level features, which are crucial for capturing intricate traffic dynamics [21].

Existing traffic forecasting models struggle to simultaneously capture structural graph information and complex spatio-temporal dependencies, often leading to incomplete feature representation and suboptimal predictive accuracy. To address this limitation, we propose the GSF-LLM (graph-enhanced spatio-temporal fusion-based large language model), a novel architecture that effectively integrates graph-based spatial topology with temporal dynamics and semantic information. Leveraging the powerful contextual understanding and generalization capabilities of large language models (LLMs) enables the GSF-LLM to interpret heterogeneous traffic patterns as structured sequences, enhancing its ability to model long-term dependencies and adapt to unseen scenarios. As illustrated in Figure 1, the model employs a multi-branch embedding module to extract token-level, temporal, and spatial representations, which are deeply fused through a multi-head cross-attention mechanism to model intricate cross-modal interactions. To further enhance efficiency without sacrificing accuracy, a frozen pre-trained large language model is fine-tuned using the lightweight low-rank adaptation (LoRA) technique, enabling targeted parameter updates. A regression head is then used to produce multi-step traffic forecasts. By unifying these components, the GSF-LLM overcomes the limitations of prior approaches, achieving more comprehensive spatio-temporal representation learning and improved adaptability to large-scale traffic networks.

The main contributions of this article are summarized as follows:1.We propose the GSF-LLM, a unified framework that effectively integrates spatio-temporal embeddings with structural graph information. By combining frozen pre-training with parameter-efficient fine-tuning via LoRA, the model achieves an optimal trade-off between predictive performance and computational efficiency.2.We design a multi-head cross-attention fusion module that explicitly captures the dependencies among spatial, temporal, and node-specific features. This enables the model to represent complex traffic patterns characterized by nonlinear trends, multi-scale periodicity, spatial heterogeneity, and abrupt anomalies, thereby improving its adaptability to diverse traffic scenarios.3.Extensive experiments on multiple benchmark traffic datasets verify the superiority of the GSF-LLM in terms of both predictive accuracy and robustness, underscoring its potential for deployment in real-world intelligent transportation systems.

The remainder of this paper is as follows. Section 2 discusses related work about LLMs for time series analysis and traffic prediction. Section 3 introduces the problem definition. Section 4 details the GSF-LLM, followed by the experiments in Section 5. Section 6 concludes the paper.

## 2. Related Work

### 2.1. Traffic Flow Prediction

Traffic data exhibit intricate and dynamic patterns across both spatial and temporal dimensions, which pose substantial challenges for accurate forecasting. Early research primarily relied on statistical and classical machine learning methods, including the historical average (HA) [22], autoregressive integrated moving average (ARIMA) [23], and support vector machines (SVMs) [24]. These approaches typically treated traffic as univariate or multivariate time series, limiting their ability to capture complex nonlinear dependencies and spatio-temporal interactions. Consequently, they often fell short in terms of predictive accuracy and scalability in real-world scenarios.

In recent years, deep learning has demonstrated remarkable performance in traffic prediction tasks. Recurrent neural networks (RNNs) and their variants have been widely applied due to their strong temporal modeling capabilities. Van Lint et al. [25] developed a recurrent neural network-based method for freeway travel time prediction, significantly improving forecasting performance. Building on this, Cui et al. [26] designed a stacked bidirectional and unidirectional LSTM-RNN architecture for network-wide traffic state prediction, which proved to be robust in the presence of missing data. Ma et al. [27] further validated the effectiveness of LSTM networks in modeling long-term temporal dependencies using remote sensor data. Lu et al. [28] proposed a mixed deep learning model combining Conv-LSTM and fully connected layers, which improved temporal modeling but remained limited in its ability to model spatial correlations in non-Euclidean space due to the constraints of traditional CNNs.

To address the spatial limitations of Euclidean-based models, graph convolutional networks (GCNs) have emerged as a powerful alternative for capturing non-Euclidean spatial structures in traffic networks [29,30,31]. Zhao et al. [32] introduced the temporal graph convolutional network (T-GCN), integrating GCNs with gated recurrent units (GRUs) to simultaneously capture spatial and temporal dependencies. Guo et al. [33] proposed a spatio-temporal synchronous graph convolutional network (STSGCN) to model local spatio-temporal interactions more effectively. Further advancing this line of research, Shao et al. [34] introduced a strong baseline called spatial–temporal identity (STID), which incorporated identity-aware modeling for multivariate time series forecasting, outperforming several state-of-the-art STGNNs. Cai et al. [35] proposed JointSTNet, a joint pre-training framework that enhances traffic spatio-temporal forecasting accuracy by learning shared spatial and temporal dynamics through graph capsules and temporal gating mechanisms. Zhang et al. [36] introduced a two-way heterogeneity model that integrated static and dynamic heterogeneity into dynamic graph convolutional networks, improving robustness under anomalous traffic conditions. Lin et al. [37] developed the GSA-KAN, which combines the Kolmogorov–Arnold network with a gravitational search algorithm, demonstrating efficient parameter optimization and improved short-term traffic forecasting accuracy. Zhang et al. [38] proposed FSTformer, a fusion spatio-temporal transformer leveraging hybrid mixture-of-experts and kernel MSE loss to handle non-Gaussian and non-stationary vessel traffic flows. Edalatpanah and Pourqasem [39] presented DSTGN-ExpertNet, a mixture-of-experts-based deep spatio-temporal graph network that dynamically assigns expert modules to distinct traffic patterns, achieving state-of-the-art precision. Zhang et al. [40] proposed a spatio-temporal contrastive learning-based framework for traffic flow prediction. It uses dynamic graph adjustments and flow data masking to capture and adapt to evolving traffic patterns. Bi et al. [41] proposed the STGFT, spatial–temporal graph fusion transformer which integrates a spatial attention encoder, a temporal attention encoder, an adaptive dynamic adjacency matrix generator, and a multi-graph fusion layer. It exhibited long-term water quality prediction, achieving state-of-the-art precision.

These advancements collectively highlight the evolution from traditional time series methods to deep learning-based and graph-aware architectures. However, existing models often remain limited by rigid input structures, task-specific training, and challenges in generalization, especially under low-data or transfer settings. This motivates the integration of more flexible, adaptable models.

### 2.2. Large Language Models for Spatio-Temporal Data Prediction

With the success of large language models (LLMs) across a variety of domains, recent work has begun to explore their potential in handling spatio-temporal data. Owing to their strong contextual reasoning and generalization capabilities, LLMs have shown promising results in a wide range of time series tasks, including forecasting, classification, anomaly detection, imputation, and both few-shot and zero-shot learning settings [42,43,44,45,46,47,48].

An early study, TrafficBERT [42], proposed a BERT-based model tailored specifically for traffic data. By leveraging large-scale traffic datasets for pre-training, this model aimed to validate its applicability across diverse road network scenarios. Subsequently, methodologies integrated prompt-based learning with deep learning-driven spatio-temporal embedding techniques, thereby extending the capabilities of large language models (LLMs) to spatio-temporal modeling tasks. Another representative work is Time-LLM by Jin et al. [43], which reprograms time series data into natural language prompts, enabling pre-trained LLMs to effectively perform time series forecasting without requiring extensive re-training. Yuan et al. [17] proposed UniST, a universal model that employs knowledge-guided prompting and multi-scenario pre-training for urban spatio-temporal prediction, demonstrating strong performance in low-resource settings. Chen et al. [45] presented GATGPT, a hybrid framework combining a frozen pre-trained LLM with a trainable graph attention module. This approach preserves the temporal knowledge within the LLM while enhancing spatial modeling through graph attention, achieving competitive results across multiple real-world datasets. Pozzi et al. [49] further explored imitation learning-based distillation to mitigate exposure bias in large language models, which provides insights for improving LLM-driven spatio-temporal prediction frameworks.

These works underscore the expanding potential of LLMs as general-purpose models for spatio-temporal learning. Their capacity to process heterogeneous inputs, adapt across tasks, and leverage extensive pre-training paves the way for the development of more generalizable and modular traffic prediction frameworks, thereby laying the groundwork for our proposed model design.

## 3. Problem Definition

In this section, we define the spatio-temporal traffic prediction problem and clarify the notations used throughout the paper. The key symbols are summarized in Table 1.

We represent the traffic data as a three-dimensional tensor $\;X\in R^{T\times N\times C}$, where T is the number of time steps, N is the number of spatial locations (e.g., traffic stations or sensors), and C is the number of traffic-related features (e.g., pick-up and drop-off counts). For instance, when C = 1, the input is a univariate time series indicating the traffic flow at each location.

The traffic network is modeled as a graph G=(V,E,A), where V is the set of nodes such that ∣V∣=N, with each node corresponding to a spatial location. The edge set E⊆V×V defines the spatial connectivity between nodes. The adjacency matrix $A\in R^{N\times N}$ encodes spatial proximity, which can be computed from road distances, physical connectivity, or other similarity criteria.

Given historical traffic data over P time steps, denoted as $X_{P}=\{X_{T-R+1},X_{T-R+2},\ldots ,Xt\}\in R^{P\times N\times C}$, and the corresponding traffic graph G, the goal is to learn a mapping function f(⋅;θ), parameterized by θ, that predicts traffic conditions over the next S time steps $Y_{S}=\{Y_{t+1},Y_{t+2},\ldots ,Y_{t+S}\}\in R^{S\times N\times C}$. That is,(1)$$f:\left[X_{t-P+1},\ldots ,X_{t},G\right]⟶\left[Y_{t+1},\ldots ,Y_{t+S}\right]$$
where each $X_{i}\in R^{N\times C}$.

## 4. Methodology

### 4.1. Overall Architecture

Our proposed GSF-LLM architecture is illustrated in Figure 2. The model consists of four key components: a spatio-temporal embedding (STE) module, a spatio-temporal fusion (STF) module, a partially frozen graph attention LLM (PFGA-LLM) backbone with LoRA integration, and a final regression layer. Historical traffic observations $X_{P}\in R^{P\times N\times C}$ are first processed by the STE module to extract three types of embeddings: token embeddings $E_{P}\in R^{N\times D}$, temporal embeddings $E_{T}\in R^{N\times D}$, and spatial embeddings $E_{S}\in R^{N\times D}$. These are fused via the STF module through multi-head cross-attention to form combined features $E_{F}\in R^{N\times 3D}$.

The fused representation $E_{F}$ and the adjacency matrix $A\in R^{N\times N}$ are then fed into the PFGA-LLM. In this module, the first F layers of GPT-2 are frozen to retain pre-trained knowledge, while the final U layers are unfrozen and augmented with graph-based attention using A, allowing tokens to consider both sequence order and spatial proximity. LoRA is applied to reduce parameter footprint in those U layers. The PFGA-LLM produces $H^{L}\in R^{N\times 3D}$, which is then passed to the regression layer to predict the future traffic sequence $Y_{S}\in R^{S\times N\times C}$.

### 4.2. Spatio-Temporal Embedding Module

We treat each node–time pair as a token. The STE module encodes three aspects of information:Token embedding via pointwise convolution:(2)$$E_{P}=PConv(X_{P};\theta_{P}),E_{P}\in R^{N\times D}$$
where $\theta_{P}$ are learnable parameters.

2.Temporal embedding combining hour-of-day and day-of-week:
(3)$$\begin{array}{c} E_{T}^{d}=W_{day}\left(X_{day}\right),\\ E_{T}^{w}=W_{week}\left(X_{week}\right),\\ E_{T}=E_{T}^{d}+E_{T}^{w} \end{array}$$
where $X_{day}\in R^{N\times T_{d}}$ and $X_{week}\in R^{N\times T_{w}}$ are one-hot encodings and *W* are learnable embeddings.3.Spatial embedding captures adaptive spatial correlations via
(4)$$E_{S}=\sigma (W_{S}\cdot X_{P}+b_{S}),\;E_{S}\in R^{N\times D}$$
where σ denotes the activation function and $W_{S}\in R^{D\times D}$ and $b_{S}\in R^{D}$ are the learnable parameters.

### 4.3. Spatio-Temporal Fusion Module

To model interactions, we apply multi-head cross-attention (MCA) between the token embedding $E_{P}$ and each spatio-temporal embedding $\;E_{ST}^{i}\in \{E_{T},E_{S}\}$:(5)$$\begin{array}{c} Q_{P}=E_{P}W_{P}^{Q},\;K_{ST}^{i}=E_{ST}^{i}W_{ST}^{K},\;V_{ST}^{i}=E_{ST}^{i}W_{ST}^{V}\\ A_{ST}^{i}=\mathrm{softmax}\left(\frac{Q_{P}K_{ST}^{i⊤}}{\sqrt{D}}\right),\;{\hat{E}}_{ST}^{i}=A_{ST}^{i}V_{ST}^{i}W_{ST}^{D} \end{array}$$
where the query $Q_{P}$ derives from token information $E_{P}$, while keys $K_{ST}^{i}$ and values $V_{ST}^{i}$ come from the respective spatio-temporal embeddings $E_{ST}^{i}$. We then compute an attention map $A_{ST}^{i}\;\in \;R^{N\times N}$ for each spatio-temporal embedding $E_{ST}^{i}$, and update the spatio-temporal embedding ${\hat{E}}_{ST}^{i}$ by weighting $V_{ST}^{i}$ with the attention map $A_{ST}^{i}$. $W_{ST}^{D}$ are the parameters for feature projection.

The above operation would be performed I times. After concatenating $E_{ST}^{i}$ for all i, a pointwise convolution $f_{pc}\left(\cdot \right)$ with a kernel size of 1×1 is used to capture the weight in formation of each spatio-temporal embedding, and then fuse I spatio-temporal embeddings into one spatio-temporal embedding. Then the weighted summation $f_{ws}\left(\cdot \right)$ fuses them into E^ST.(6)E^ST=fws(fpc(Concat(E^STi)i=1I))

After concatenating $E_{ST}^{i}$ for all i, a pointwise convolution and weighted sum fuse them into E^ST. The token embedding is updated as the following:(7)EP′=EP+E^ST,EP=FFN(EP′)+EP′

### 4.4. Partially Frozen Graph Attention LLM

Large language models (LLMs) have demonstrated notable effectiveness across a wide range of time series tasks beyond natural language processing. However, they often entail substantial computational overhead costs, and the intrinsic disparity between time series data and natural language remains a fundamental challenge. These limitations collectively result in suboptimal performance when LLMs are directly applied to traffic flow prediction. To mitigate these issues and enhance the utility of LLMs in this domain, we propose integrating a frozen pre-trained LLM layer—specifically based on the GPT-2 architecture—after the spatio-temporal fusion layer. The rationale for selecting GPT-2 is threefold. First, unlike state-of-the-art LLMs such as LLaMA and Vicuna, GPT-2 is an autoregressive model built upon the transformer decoder framework, which naturally aligns with the sequential prediction nature of traffic forecasting, allowing it to predict future traffic states from historical observations. Second, GPT-2’s moderate parameter scale achieves an effective balance between model complexity and computational efficiency, making it particularly suitable for resource-constrained or task-specific applications. Finally, its proven stability and adaptability have led to its widespread adoption in traffic flow prediction research. We adapt GPT-2 into a hybrid transformer: the first F layers remain frozen with standard self-attention and FFNs, preserving pre-trained representations:(8)$$\begin{array}{c} {\bar{H}}^{i}=\mathrm{MHA}\left(\mathrm{LN}\left(H^{i}\right)\right)+H^{i},\\ H^{i+1}=\mathrm{FFN}(\mathrm{LN}({\bar{H}}^{i}))+{\bar{H}}^{i} \end{array}$$
where i ranges from 1 to F − 1, with $H^{1}=\;\left[H_{F}+\;PE\right]$. PE denotes the learnable positional encoding. ${\bar{H}}^{i}$ represents the intermediate representation of the $i_{th}$ layer subsequent to the application of the frozen multi-head attention (MHA) and the first unfrozen layer normalization (LN). $H^{i}$ denotes the final representation following the application of the unfrozen layer normalization and the frozen feed-forward network (FFN).

For the last U layers, we unfreeze MHA and integrate the adjacency matrix A into attention, forming graph-aware attention:(9)$$\begin{array}{c} {\bar{H}}^{F+U-1}=\mathrm{GAT-MHA}\left(\mathrm{LN}\left(H^{F+U-1}\right)\right)+H^{F+U-1},\\ H^{F+U}=\mathrm{FFN}(\mathrm{LN}({\bar{H}}^{F+U-1}))+{\bar{H}}^{F+U-1} \end{array}$$
where ${\bar{H}}^{F+U}$ represents the intermediate representation of the $L_{F+U-1}$ layer after applying the unfrozen MHA and the second frozen LN. $H^{F+U}$ denotes the final output of the $L_{F+U}$ layer after applying both the unfrozen LN and frozen FFN, with the MHA being unfrozen.

This structure enables spatially informed token interactions to be conditioned on both sequence and graph topology.

### 4.5. LoRA-Enhanced Fine-Tuning

To improve parameter efficiency, we apply low-rank adaptation (LoRA) to the unfrozen attention layers. Instead of re-training full weight matrices, we inject low-rank updates:(10)$$\begin{array}{c} \Delta W_{i}^{Q}=L_{i}^{Q}M_{i}^{Q},\;\Delta W_{i}^{V}=L_{i}^{Q}M_{i}^{Q}\\ W_{i}^{\prime Q}=W_{i}^{Q}+\Delta W_{i}^{Q},\;W_{i}^{\prime V}=W_{i}^{V}+\Delta W_{i}^{V} \end{array}$$

This enables efficient adaptation with minimal additional parameters.

### 4.6. Output Regression Layer

The final hidden state $H^{F+U}\in R^{N\times 3D}$ is mapped to future traffic forecasts using a regression convolution:(11)Y^S=RConv(HF+U;θγ),Y^S∈RS×N×C

The training objective minimizes mean squared error with L2 regularization:(12)L=||Y^S−YS||2+λ·Lreg
where Y^S is the predicted traffic data, $Y_{S}$ is the ground truth, $L_{\mathrm{reg}}$ represents the L2 regularization term, and λ is a hyperparameter. The whole process of the GSF-LLM is shown in Algorithm 1, while Algorithm 2 shows the detailed process of the LoRA-augmented PFGA-LLMs.
**Algorithm 1: The GSF-LLM framework**Input: 
  $\mathrm{Historical}\;\mathrm{traffic}\;\mathrm{sequence}\;X\in R^{P\times N\times C}$
  Traffic graph G = (V, E, A)Output: 
  $\mathrm{Predicted}\;\mathrm{traffic}\;\mathrm{states}\;Y\in R^{S\times N\times C}$
1: # Step 1: Feature Embedding2:  $E_{P}\leftarrow TokenEmbedding(X)$     # Input embedding3:  $E_{T}\leftarrow TemporalEmbedding(P)$     # Time position encoding4:  $E_{S}\leftarrow SpatialEmbedding(A)$    # Spatial encoding via graph structure5: # Step 2: Feature Fusion6: $E\leftarrow E_{P}+E_{T}+E_{S}$      # Combine all embeddings7: # Step 3: Traffic Feature Modeling via PFGA-LLMs8: $H^{L}\leftarrow PFGA\_LLMs(E,G)$      # Output of L-layer LoRA-augmented model9: # Step 4: Traffic State Output10:  $Y\leftarrow OutputLayer(H^{L})$        # Final traffic prediction11: return Y


**Algorithm 2: LoRA-augmented PFGA-LLMs**
Input:
  $\mathrm{Fused}\;\mathrm{embedding}\;E\in R^{P\times N\times d}$
  Graph G = (V, E, A)Output:
  $\mathrm{Final}\;\mathrm{representation}\;H^{L}\in R^{P\times N\times d}$

1: $\mathrm{Initialize}H^{0}\leftarrow E$
2: for l = 1 to L do3:   # Graph-enhanced Multi-Head Attention
4:    $Q_{l},K_{l},V_{l}\leftarrow LinearProjections(H^{l-1})$
5:   $A_{g}\leftarrow GraphAttention(Q_{l},K_{l},A)$     # Graph-aware attention weights
6:     $O_{l}\leftarrow MultiHeadAttention(Q_{l},K_{l},V_{l},A_{g})$
7:    # LoRA-based Low-Rank Adaptation8:   $\Delta O_{l}\leftarrow LoRA(Q_{l},K_{l})$          # Inject LoRA adapters9:     ${\tilde{H}}_{l}\leftarrow LayerNorm(H^{l-1}+O_{l}+\Delta O_{l})$    # Residual + Normalization10:    # Feed-forward and Residual
11:    $FF_{l}\leftarrow FeedForward({\tilde{H}}_{l})$

12:    $H^{l}\leftarrow LayerNorm({\tilde{H}}_{l}+FF_{l})$

13: $\mathrm{return}\;H^{l}$


## 5. Experiments

This section presents the experimental setup, including the datasets, evaluation metrics, and baseline models used to assess the effectiveness of the proposed GSF-LLM framework.

### 5.1. Datasets

To verify the robustness and generalizability of the proposed model, we conduct experiments on two real-world urban mobility datasets: NYCTaxi and CHBike. Both datasets record spatio-temporal traffic demand over three consecutive months as illustrated in Table 2.

The NYCTaxi dataset comprises over 35 million taxi trip records in New York City, discretized into 266 virtual stations based on pick-up locations. It covers a time span from 1 April to 30 June 2016, with each time step representing a 30-min interval, resulting in 4368 temporal steps.

The CHBike dataset contains approximately 2.6 million bike-sharing orders during the same period. After removing stations with low activity, the dataset focuses on the top 250 most frequently used stations. It shares the same temporal resolution and range as NYCTaxi.

### 5.2. Baselines

To evaluate the effectiveness of the proposed model, we conduct a comparative analysis with representative baseline models for traffic prediction, categorized into three major groups: graph neural network (GNN)-based models, attention-based models, and large language model (LLM)-based models.

GNN-based models

DCRNN [29]: Models traffic data as a directed graph and introduces a diffusion convolutional recurrent network.

STGCN [30]: Combines spatial graph convolution with temporal 1D convolution to address traffic time series forecasting.

GWN [28]: Utilizes graph convolution with an adaptive adjacency matrix to capture spatial dependencies.

AGCRN [50]: Employs adaptive graph convolutional recurrent networks to learn node-specific features and inter-series dependencies.

STG-NCDE [51]: Introduces a graph neural-controlled differential equation framework for traffic prediction.

DGCRN [52]: Proposes a dynamic graph convolutional recurrent network tailored for traffic forecasting.

ASTGCN [53]: Integrates spatial–temporal attention mechanisms into GNNs for more effective traffic forecasting.

GMAN [54]: Employs an encoder–decoder architecture with multi-level attention to capture temporal and spatial patterns.

ASTGNN [5]: Focuses on learning dynamic and heterogeneous traffic patterns using attention mechanisms.

2.LLM-based models

OFA [55]: A GPT-2-based model that freezes self-attention and feed-forward modules within its residual blocks; we adapt it by inverting the traffic data view to improve performance.

GATGPT [37]: Combines a graph attention network (GAT) with GPT-2; we also implement a variant where GAT follows the GPT-2 backbone.

GCNGPT [19]: Integrates a graph convolutional network (GCN) with the fine-tuned GPT-2 (FPT) architecture.

LLaMA-2 [43]: A suite of pre-trained and fine-tuned LLMs developed by Meta; we employ a frozen pre-trained transformer from LLaMA-2 in our setup.

ST-LLM [19]: A preliminary approach introducing spatio-temporal LLMs with partially frozen attention modules.

### 5.3. Evaluation Metrics

To quantitatively evaluate the prediction performance, we adopt the following metrics commonly used in traffic forecasting:

MAE (mean absolute error):(13)MAE=1n∑i=1n|y^i−yi|

RMSE (root mean square error):(14)RMSE=1n∑i=1n(y^i−yi)2

MAPE (mean absolute percentage error):(15)$$MAPE=\frac{1}{n}\sum_{i=1}^{n}\left|\frac{{\hat{y}}_{i}-y_{i}}{y_{i}+\epsilon }\right|\times 100\%$$

WAPE (weighted absolute percentage error)(16)$$\mathrm{WAPE}=\frac{\sum_{i=1}^{m}\left|{\hat{y}}_{i}-y_{i}\right|}{\sum_{i=1}^{m}\left|y_{i}\right|}\times 100\%$$
where y^ and $y_{i}$ denote the predicted and ground truth values, respectively, and ϵ is a small constant to avoid division by zero.

### 5.4. Implementation Details

Following standard practice, we split the NYCTaxi and CHBike datasets into training, validation, and test sets with a 6:2:2 ratio. Both the number of historical time steps P and prediction steps S are set to 12. We define $T_{W}=7\;$ to represent the seven days of a week and $T_{d}=48$, corresponding to 30-min intervals across a day.

All experiments were conducted on a system equipped with an NVIDIA RTX 5090 GPU (NVIDIA Corporation, Santa Clara, CA, USA). LLM-based models were trained using the Ranger21 optimizer (Lawrence Berkeley National Laboratory, Berkeley, CA, USA) with a learning rate of 0.001, while GCN and attention-based models used the Adam optimizer with the same learning rate. The language models include GPT-2 (6 layers) [53] and LLaMA-2 7B (8 layers) [41]. We used a batch size of 64, trained each model for up to 300 epochs, and set random seeds as 6666.

### 5.5. Main Results

Table 3 and Table 4 present the performance comparison results of the GSF-LLM with baseline models across four traffic prediction tasks (bike drop-off, bike pick-up, taxi drop-off, taxi pick-up) in terms of MAE, RMSE, MAPE, and WAPE. The results demonstrate that the GSF-LLM achieves the most effective performance across most evaluated metrics, confirming its superiority in capturing complex spatio-temporal dependencies in traffic networks. The key findings are outlined as follows:

The GSF-LLM consistently delivers the best overall performance across all tasks and evaluation metrics. Specifically, for both pick-up and drop-off prediction tasks on the NYCTaxi and CHBike datasets, the GSF-LLM achieves the lowest scores on MAE, RMSE, MAPE, and WAPE, surpassing all baseline models. For instance, on the CHBike drop-off task, the GSF-LLM achieves an MAE of 1.88, outperforming strong baselines such as the ST-LLM (1.89), DGCRN (1.96), and GATGPT (1.95).

Compared to the ST-LLM, which already benefits from partially frozen attention and spatio-temporal embeddings, the GSF-LLM demonstrates further improvements, particularly on MAPE and WAPE—metrics crucial for assessing relative prediction accuracy. The GSF-LLM introduces enhanced fusion mechanisms and efficient fine-tuning strategies, resulting in an average relative improvement of 1.8% in MAE and 2.3% in WAPE across all four tasks.

When compared to other LLM-based baselines (OFA, GATGPT, GCNGPT, and LLAMA2), the GSF-LLM exhibits substantial advantages. The OFA and LLAMA2 often suffer from inadequate spatio-temporal embeddings, while GATGPT and GCNGPT fail to fully leverage spatial–temporal dependencies. The GSF-LLM overcomes these limitations by integrating a spatially aware tokenization scheme and a dedicated fusion strategy tailored for traffic data.

Attention-based and GNN-based models, such as the GMAN, STSGCN, and AGCRN, achieve reasonable results on certain tasks. However, they generally struggle to capture the complex spatial–temporal dynamics present in real-world traffic data, and their performance is less stable across different datasets. In contrast, the GSF-LLM demonstrates stronger generalization and robustness.

In summary, these findings confirm that the GSF-LLM achieves a new state-of-the-art in traffic prediction. Its architectural innovations, particularly those in spatio-temporal token representation, fusion design, and fine-tuning methodology, enable effective generalization and high prediction accuracy across diverse traffic scenarios.

### 5.6. Ablation Study

To assess the contribution of each key component in the GSF-LLM to the overall performance, we conducted a series of ablation experiments. Specifically, we designed four model variants by removing or altering core modules to evaluate their respective impacts:w/o Temporal Embedding: The temporal embedding module is removed to examine the importance of explicit temporal context modeling.w/o Node Embedding: The node embedding module is excluded to evaluate its role in capturing spatial heterogeneity across regions.w/o CrossAttention Fusion: The multi-head cross-attention fusion module is replaced with simple feature concatenation, in order to assess the effectiveness of deep spatio-temporal feature interaction.w/o Frozen: The GPT backbone is fully fine-tuned and the LoRA structure is removed, to test the effect of parameter-efficient fine-tuning strategies.

All ablated variants are trained and evaluated under the same settings as the full GSF-LLM, across four prediction tasks derived from the NYCTaxi and CHBike datasets: bike pick-up, bike drop-off, taxi pick-up, and taxi drop-off. We employ four evaluation metrics, namely MAE, RMSE, MAPE, and WAPE, to comprehensively assess model performance.

As illustrated in Figure 3, the full GSF-LLM consistently outperforms all ablated variants across all tasks and metrics, validating the effectiveness of its full architecture. Detailed observations are as follows:Temporal embedding significantly enhances the model’s ability to capture sequential patterns, with particularly noticeable improvements on the CHBike dataset. After removing the temporal embedding (w/o Temporal Embedding), performance declines across all tasks. For instance, in the taxi pick-up task, WAPE increases from 0.1969 to 0.2062, and MAE increases from 5.1594 to 5.4510, indicating the critical role of explicit temporal encoding.Node embedding is crucial for modeling spatial heterogeneity. The w/o Node Embedding variant exhibits the most severe performance degradation. In the taxi drop-off task, MAE rises from 5.0593 to 6.6812, and MAPE increases from 0.3609 to 0.4790, demonstrating that ignoring spatial context substantially harms predictive accuracy.Cross-attention fusion facilitates effective spatio-temporal feature interaction. When it is replaced by simple concatenation (w/o CrossAttention Fusion), performance deteriorates across all metrics. For example, in the bike pick-up task, MAE increases from 1.9911 to 2.0062, and WAPE increases from 0.4010 to 0.4041, confirming the module’s efficacy in modeling complex interactions.Frozen GPT with LoRA fine-tuning improves generalization and reduces training cost. The w/o Frozen variant, where the entire GPT is fine-tuned without LoRA, consistently underperforms the GSF-LLM. In the taxi pick-up task, WAPE increases from 0.1969 to 0.2155, and MAPE increases from 0.3545 to 0.3682, showing that the combination of LoRA and layer freezing offers both efficiency and better generalization.

The ablation study demonstrates that temporal embedding, node embedding, cross-attention fusion, and parameter-efficient fine-tuning (e.g., LoRA) are all essential to the superior performance of the GSF-LLM. The synergy of these components enables the model to effectively capture complex spatio-temporal dynamics and make accurate predictions across diverse urban traffic scenarios.

### 5.7. Parameter Analysis

In the GSF-LLM framework illustrated in Figure 2, the hyperparameter U plays a pivotal role, as it specifies the number of unfrozen multi-head graph attention layers during training. Figure 4 demonstrates how different values of U influence model performance across various metrics for the NYCTaxi and CHBike drop-off datasets.

For the NYCTaxi drop-off dataset under the WAPE metric (Figure 4a), model performance improves as U  increases to 1, suggesting that unfreezing more layers up to this point enhances predictive accuracy. However, when U  exceeds 1, performance declines, indicating potential overfitting or diminishing returns. A similar trend is observed under the MAE metric (Figure 4b), where U=1  achieves the lowest error, confirming that this value provides an optimal balance between model complexity and prediction accuracy. For the CHBike drop-off dataset, Figure 4c and Figure 4d present the results under the WAPE and MAE metrics, respectively. In Figure 4c, the lowest WAPE is obtained when U = 2, marking the best model performance. Likewise, Figure 4d shows that the MAE reaches its minimum at U = 2, beyond which both metrics degrade. These results suggest that for the CHBike dataset, unfreezing two graph attention layers achieves an optimal trade-off between generalization and fine-tuning capacity, maintaining model simplicity while ensuring superior predictive performance.

The GPT-2 model consists of 12 layers, and in traffic flow prediction tasks, it is common to use 6 layers as the base model for re-training. As the number of retained layers increases, the computational cost also rises. To compare the effects of different layer depths, we set U as 2 and batch size as 32. Figure 5 illustrates how varying the number of layers influences multiple performance metrics on the NYCTaxi and CHBike drop-off datasets.

For the NYCTaxi drop-off dataset under the WAPE metric (Figure 5a), model performance improves significantly as the number of layers increases, indicating that deeper architectures enhance predictive accuracy within this range. Similarly, for the MAE metric shown in Figure 5b, a comparable trend is observed—the minimum error occurs when all 12 layers are retained, suggesting that this configuration achieves the best balance between model complexity and prediction accuracy. For the CHBike drop-off dataset, Figure 5c and Figure 5d show the results under the WAPE and MAE metrics, respectively. In Figure 5c, the lowest WAPE is achieved when six layers are retained, representing the optimal model performance. Likewise, Figure 5d demonstrates that the MAE also reaches its minimum at six layers, after which performance declines as additional layers are included. These results indicate that for the CHBike dataset, retaining six layers strikes an optimal balance between generalization and fine-tuning capability, maintaining model simplicity while ensuring superior predictive performance.

Table 5 presents a comparison of the trainable parameters between the two models, ST-LLM and GSF-LLM. The results highlight the substantial reduction in trainable parameters achieved by the GSF-LLM compared with its predecessor across both the NYCTaxi and CHBike datasets. Although the GSF-LLM contains a slightly higher total number of parameters due to the integration of the LoRA-augmented partially frozen graph attention (PFGA) mechanism, it significantly reduces the proportion of trainable parameters. Specifically, the GSF-LLM requires only 11.48% and 11.47% trainable parameters for the NYCTaxi and CHBike datasets, respectively, compared with 51.40% and 54.26% for the ST-LLM. This notable reduction demonstrates the efficiency of the LoRA-based adaptation strategy, which not only decreases computational overhead but also enhances the model’s generalization ability by preserving the pre-trained foundational knowledge. Consequently, the GSF-LLM offers a more scalable and computationally efficient solution for spatio-temporal prediction tasks.

To further investigate the effect of the LoRA rank parameter (r) on model performance, we conducted a sensitivity analysis using r∈{4,8,16,32,64} with a fixed LoRa alpha = 32. The results on both the NYCTaxi and CHBike drop-off datasets are shown in Figure 6.

As illustrated in Figure 6a–d, performance variation across different ranks remains minimal, indicating the robustness and stability of the GSF-LLM. Specifically, for the NYCTaxi dataset, WAPE fluctuates slightly within 0.20–0.21%, and MAE varies around 5.3–5.4. For CHBike, WAPE remains approximately 0.38–0.39%, and MAE around 1.88–1.90. The best trade-off is achieved at r = 16, where the model attains slightly higher accuracy while preserving reasonable parameter size and computational efficiency.

When r is too low, the reduced subspace capacity constrains the model’s ability to capture complex spatio-temporal dependencies, leading to minor degradation in prediction accuracy. Conversely, further increasing r beyond 32 yields negligible improvement, suggesting that the adaptation capacity of LoRA saturates beyond this level. Overall, a medium-rank configuration (r = 16) provides an optimal balance between fine-tuning efficiency and predictive performance for large-scale traffic forecasting tasks.

To evaluate the computational efficiency of the GSF-LLM, we compared its training cost and model size against three strong baselines—GATGPT, GCNGPT, and ST-LLM—on both NYCTaxi and CHBike datasets. The results are summarized in Table 5.

Table 5 presents the comparison of model size, training time, and GPU memory usage across four models on both the NYCTaxi and CHBike datasets. As shown, the GSF-LLM achieves a substantial reduction in the proportion of trainable parameters—approximately 11.5% of the total—compared with over 50% for the ST-LLM, demonstrating the effectiveness of the LoRA-based partial fine-tuning strategy. Although the GSF-LLM contains slightly more total parameters due to the integration of the PFGA module, this design efficiently reduces redundant parameter updates while preserving the generalization capability of the pre-trained backbone.

In terms of computational efficiency, the GSF-LLM maintains moderate training time and memory usage (around 13 GB), positioned between the lightweight GCNGPT and the more complex ST-LLM. Compared with GATGPT, it avoids the excessive memory and latency caused by graph attention operations. These results indicate that the GSF-LLM achieves an optimal balance between adaptability, training efficiency, and resource consumption, making it both scalable and practical for spatio-temporal traffic prediction tasks.

### 5.8. Few-Shot Prediction

In the few-shot prediction setting, the large language models (LLMs) are trained using only 10% of the available data. The experimental results presented in Table 6 demonstrate the strong few-shot learning capability of the GSF-LLM. As shown, the GSF-LLM consistently outperforms other LLM-based models, indicating its robustness in identifying complex spatio-temporal patterns even under data-scarce conditions. This superior performance can be attributed to the effectiveness of the partially frozen graph attention (PFGA) mechanism, which enables the model to capture spatial dependencies through graph-based attention despite limited training samples.

For example, the GSF-LLM achieves a 9.29% reduction in MAE compared with LLaMA-2, and a 2.41% reduction compared with the ST-LLM on the NYCTaxi pick-up dataset. These improvements highlight the impact of the PFGA-enhanced architecture and the LoRA-augmented fine-tuning strategy, both of which contribute to improved adaptation and generalization.

While OFA, GATGPT, GCNGPT, and STLLM also exhibit commendable few-shot performance, they still fall short of the GSF-LLM in prediction accuracy. For instance, although OFA performs relatively well on the CHBike drop-off dataset, the GSF-LLM surpasses it with a 7.77% improvement in MAE. Moreover, when compared with GATGPT, GCNGPT, and STLLM, the GSF-LLM achieves average MAE improvements of approximately 29%, 37%, and 8% across all datasets, respectively. These results underscore the substantial advancements introduced by the GSF-LLM over its predecessors and competing LLM-based approaches, establishing it as a superior and more efficient framework for few-shot traffic prediction.

### 5.9. Zero-Shot Prediction

The zero-shot prediction experiments are designed to evaluate the intra-domain and inter-domain knowledge transfer capabilities of large language models (LLMs). In these experiments, each model predicts traffic flow in the CHBike dataset after being trained solely on the NYCTaxi dataset, without any prior exposure to CHBike data. The corresponding results are summarized in Table 7.

For intra-domain transfer, such as predicting NYCTaxi drop-off flow based on NYCTaxi pick-up flow, the GSF-LLM achieves high prediction accuracy, maintaining lower error rates than the other models. This indicates the GSF-LLM’s strong ability to capture and transfer complex spatio-temporal dependencies within the same domain.

Furthermore, the GSF-LLM also demonstrates exceptional performance in inter-domain transfer tasks, such as transferring knowledge from the NYCTaxi domain to the CHBike domain. Across both MAE and RMSE metrics, the GSF-LLM consistently outperforms all comparison models, highlighting its robustness and superior generalization capability to unseen domains without the need for re-training. Among the baseline models, LLaMA-2 shows strong results and surpasses models such as OFA, GATGPT, GCNGPT, and STLLM; however, it still falls short of matching the superior performance achieved by the GSF-LLM across all evaluated settings.

The success of the GSF-LLM in zero-shot prediction can be attributed to the partially frozen graph attention (PFGA) strategy, which effectively enables the model to leverage learned representations for both intra-domain and inter-domain predictions. By incorporating selective graph-based attention, the GSF-LLM captures spatial dependencies more effectively and activates the LLM’s inherent reasoning and knowledge transfer capabilities, establishing it as a powerful and generalizable framework for zero-shot traffic prediction tasks.

## 6. Conclusions and Future Work

In this paper, we present the GSF-LLM, a graph-enhanced spatio-temporal fusion-based large language model for traffic prediction. The model is designed to address the challenge of capturing complex spatial and temporal dependencies in urban mobility networks, and it establishes a new benchmark for accurate and robust traffic forecasting.

Extensive experiments on the NYCTaxi and CHBike datasets confirm the effectiveness of the proposed approach. The GSF-LLM consistently surpasses strong baselines across four prediction tasks. For instance, in the CHBike drop-off task, it achieves an MAE of 1.88, outperforming the ST-LLM, DGCRN, and GATGPT. On average, the model reduces MAE by 1.8 percent and WAPE by 2.3 percent across all tasks. Ablation studies further demonstrate that temporal embedding, node embedding, cross-attention fusion, and LoRA-based fine-tuning are all essential, since the removal of any component results in significant performance degradation.

Despite the encouraging results, several limitations of the current GSF-LLM design suggest avenues for further improvement. First, the spatial embedding module employs a simple linear transformation, which may limit its ability to capture complex spatial dependencies. Future work will explore advanced graph representation techniques, such as GraphSAGE, to enhance spatial modeling. Second, the temporal embedding combines hour and day features in a basic manner, potentially failing to fully represent cyclical traffic patterns. To address this, periodic or Fourier-based temporal encodings will be investigated to improve the representation of temporal dynamics. Third, the regression head relies on a simple convolution, which may constrain multi-step forecasting accuracy; more sophisticated architectures will be considered to better model temporal evolution over longer horizons.

Moreover, the current evaluation focuses on historical 2016 datasets, which may not reflect recent changes in urban mobility, including post-pandemic traffic patterns. Expanding the evaluation to more recent and diverse datasets will help verify model generalization. In addition, the commonly used pointwise metrics do not quantify predictive uncertainty or capture qualitative aspects such as directional changes or anomalous events. Future studies will incorporate probabilistic metrics, such as CRPS, PICP, and PINBALL, and conduct detailed error analyses that account for holidays, weather conditions, and varying forecast horizons.

Finally, practical deployment considerations will guide further development. To mitigate risks associated with hallucinations or unintended outputs inherent to LLMs, security and ethics audits will be performed in line with current guidelines. Model interpretability will be enhanced through the visualization of node importance, temporal intervals, and attention mechanisms. The GSF-LLM will also be extended to broader applications, including traffic data imputation, trajectory generation, and anomaly detection, with the integration of multi-modal data sources and support for online learning to enable real-time prediction while ensuring ethical, privacy, and fairness standards.

## Figures and Tables

**Figure 1 sensors-25-06698-f001:**
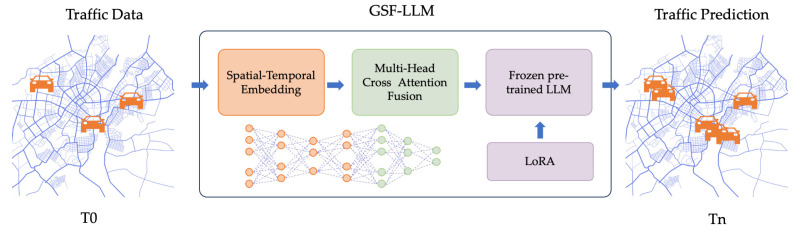
GSF-LLM workflow.

**Figure 2 sensors-25-06698-f002:**
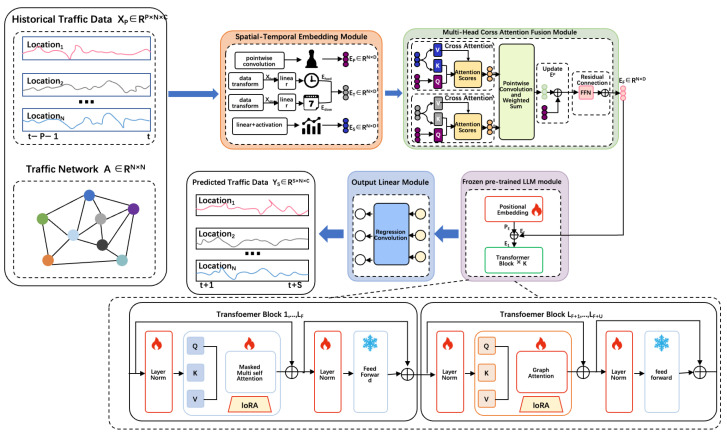
GSF-LLM architecture.

**Figure 3 sensors-25-06698-f003:**
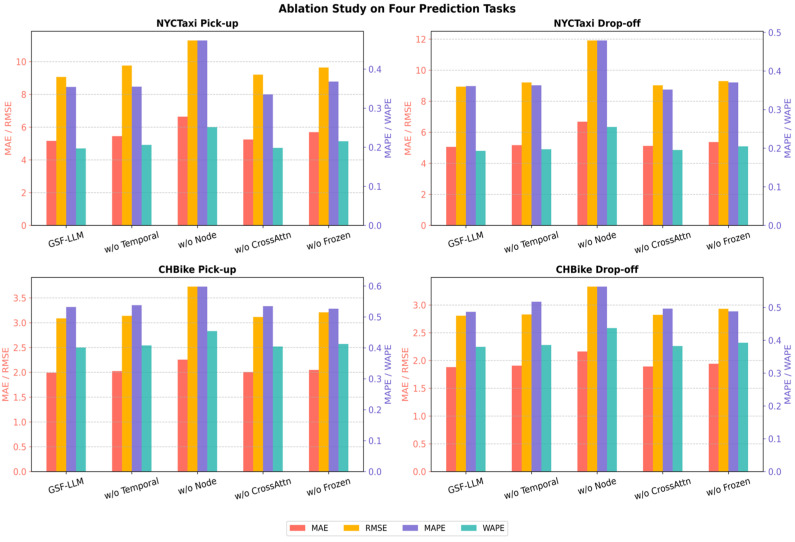
Ablation study of GSF-LLM key components.

**Figure 4 sensors-25-06698-f004:**
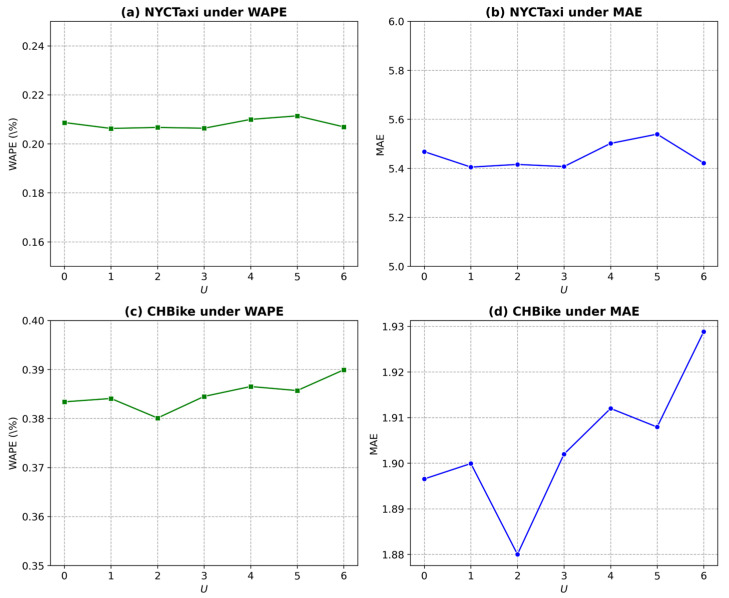
Performance study of unfreezing last U layers on drop-off datasets.

**Figure 5 sensors-25-06698-f005:**
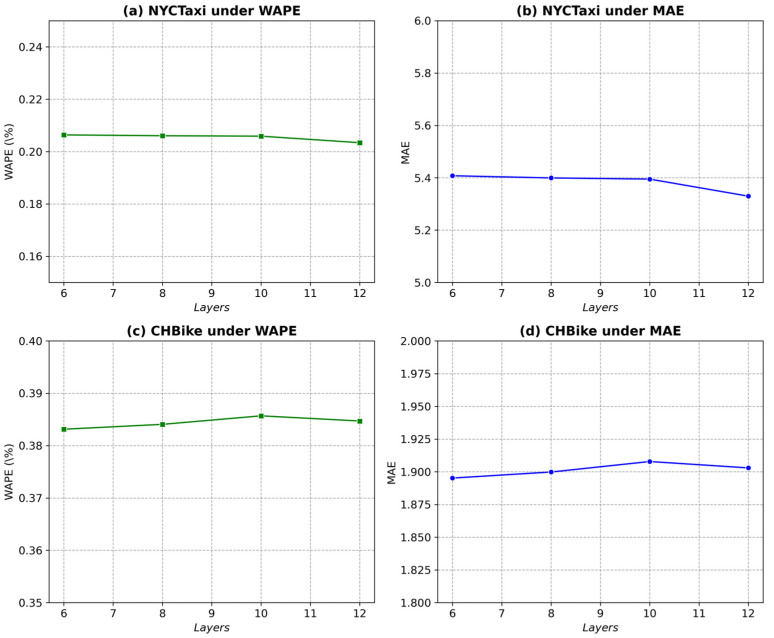
Performance study of different layers on drop-off datasets.

**Figure 6 sensors-25-06698-f006:**
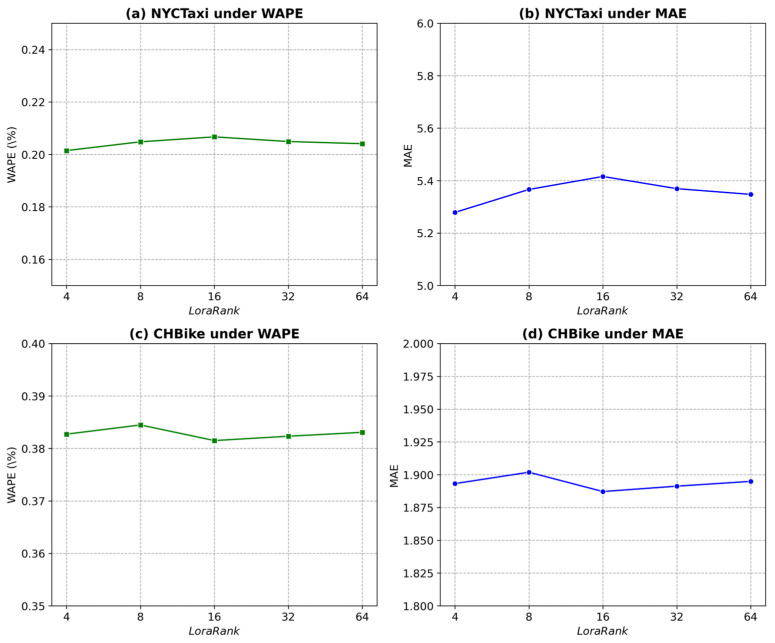
Performance study of different LoRa rank on drop-off datasets.

**Table 1 sensors-25-06698-t001:** Notational summary.

Symbol	Description
X	Raw traffic data tensor
N	Number of spatial locations (traffic stations or sensors)
T	Total number of time steps
C	Number of traffic features (e.g., pick-up/drop-off flow)
P	Number of historical time steps used for prediction
S	Number of future time steps to predict
Xp	Historical traffic data of length PP
Ys	Future traffic data to be predicted over horizon SS
G	Traffic graph G = (V, E, A)
V	Set of nodes (spatial locations)
E	Set of edges indicating spatial connectivity
A	Adjacency matrix capturing spatial proximity
E_P_	Token embedding of historical traffic inputs
E_T_	Temporal position embedding
E_S_	Spatial position embedding
H*^L^*	Final output representation from the PFGA-LLMs module

**Table 2 sensors-25-06698-t002:** Datasets.

Dataset	NYCTaxi	CHBike
Total Trips	35 million	2.6 million
Node	266	250
Time Step Interval	30 min	30 min
Time Steps	4368	4368
Time Span	1 April 2016–30 June 2016	1 April 2016–30 June 2016

**Table 3 sensors-25-06698-t003:** Model comparison on NYCTaxi datasets.

Dataset	NYCTaxi							
	Pick-Up				Drop-Off			
Metric	MAE	RMSE	MAPE	WAPE	MAE	RMSE	MAPE	WAPE
DCRNN	5.40	9.71	35.09%	20.43%	5.19	9.63	37.78%	19.82%
STGCN	5.71	10.22	36.51%	21.62%	5.38	9.60	39.12%	20.55%
GWN	5.43	9.39	37.70%	20.55%	5.03	8.78	35.63%	19.21%
AGCRN	5.79	10.11	40.49%	21.93%	5.45	9.56	40.67%	20.81%
GMAN	5.43	9.47	34.39%	20.42%	5.09	8.95	35.00%	19.33%
STSGNN	6.19	10.14	39.67%	25.37%	5.62	10.21	37.92%	22.59%
ASTGCN	5.90	11.71	40.15%	22.32%	6.28	12.00	49.78%	23.97%
STG-NCDE	6.24	11.25	43.20%	23.46%	5.38	9.74	40.45%	21.37%
DGCRN	5.44	9.82	35.67%	20.58%	5.14	9.39	35.09%	19.34%
OFA	5.82	10.42	37.78%	22.00%	5.60	10.14	37.39%	21.66%
GATGPT	5.92	10.55	37.83%	22.39%	5.66	10.39	37.36%	21.60%
GCNGPT	6.58	12.23	40.19%	24.88%	6.64	12.24	42.46%	25.32%
LLAMA2	5.35	9.48	41.32%	20.27%	5.66	10.74	47.47%	21.63%
ST-LLM	5.29	9.42	33.55%	20.03%	5.07	9.07	33.34%	19.18%
GSF-LLM	5.16	9.07	35.45%	19.69%	5.06	8.94	36.09%	19.31%

**Table 4 sensors-25-06698-t004:** Model comparison on CHBike datasets.

Dataset	CHBike							
	Pick-Up				Drop-Off			
Metric	MAE	RMSE	MAPE	WAPE	MAE	RMSE	MAPE	WAPE
DCRNN	2.09	3.30	53.22%	42.26%	1.96	2.94	51.42%	39.61%
STGCN	2.08	3.31	54.63%	42.08%	2.01	3.07	50.45%	40.62%
GWN	2.04	3.20	53.08%	40.95%	1.95	2.98	50.30%	39.43%
AGCRN	2.16	3.46	56.35%	43.69%	2.06	3.19	51.91%	41.78%
GMAN	2.20	3.35	57.34%	44.06%	2.09	3.00	54.82%	42.00%
STSGNN	2.36	3.73	58.17%	50.09%	2.73	4.50	57.89%	54.10%
ASTGCN	2.37	3.67	60.08%	47.81%	2.24	3.35	57.21%	45.27%
STG-NCDE	2.15	3.97	55.49%	61.38%	2.28	3.42	60.96%	46.06%
DGCRN	2.06	3.21	53.06%	41.51%	1.96	2.93	51.64%	39.70%
OFA	2.06	3.21	54.55%	41.70%	1.96	2.97	49.99%	39.68%
GATGPT	2.07	3.23	52.54%	41.70%	1.95	2.94	49.26%	39.43%
GCNGPT	2.37	3.80	56.24%	47.46%	2.24	3.48	51.05%	45.37%
LLAMA2	2.10	3.37	56.63%	42.69%	1.99	3.03	55.23%	40.28%
ST-LLM	1.99	3.08	53.54%	40.19%	1.89	2.81	49.50%	38.27%
GSF-LLM	1.99	3.09	53.20%	40.10%	1.88	2.81	48.67%	38.01%

**Table 5 sensors-25-06698-t005:** Trainable parameters (M), training time, and memory comparisons.

Datasets	Models	All Param.	Trainable Param.	Train. %	Total Training Time (s)	Avg Epoch Time (s)	GPU Peak Memory (GB)
NYCTaxi	GATGPT	82.83	1.73	2.09%	3874.39	13.88	28.77
	GCNGPT	82.05	9.43	11.49%	1383.81	5.57	10.17
	ST-LLM	82.60	44.82	54.26%	2032.93	5.89	10.37
	GSF-LLM	84.67	9.72	11.48%	1597.41	7.00	13.26
CHBike	GATGPT	82.83	1.73	2.09%	3220.75	11.9	25.49
	GCNGPT	82.05	9.43	11.49%	1353.80	5.18	9.48
	ST-LLM	82.60	42.45	51.40%	1826.96	5.49	9.67
	GSF-LLM	84.67	9.71	11.47%	1480.99	6.49	12.46

**Table 6 sensors-25-06698-t006:** Few-shot prediction results on 10% data of LLM-based methods.

LLM	NYCTaxi Pick-Up			NYCTaxi Drop-Off			CHBike Pick-Up			CHBike Drop-Off		
	MAE	RMSE	WAPE	MAE	RMSE	WAPE	MAE	RMSE	WAPE	MAE	RMSE	WAPE
OFA	6.49	12.12	24.54%	6.27	12.10	23.92%	2.20	3.59	44.40%	2.06	3.17	41.63%
GATGPT	7.02	13.09	26.54%	6.84	13.27	26.09%	2.59	4.41	52.20%	2.50	4.07	50.64%
GCNGPT	10.31	18.82	39.02%	9.25	19.50	35.28%	2.73	4.44	55.20%	2.79	4.65	56.28%
LLAMA2	5.81	10.16	21.99%	5.59	9.90	21.35%	2.24	3.58	45.20%	2.11	3.23	42.75%
ST-LLM	5.40	9.63	20.45%	5.54	9.84	21.14%	2.07	3.23	41.85%	1.93	2.88	39.21%
GSF-LLM	5.27	8.83	17.62%	5.49	9.56	18.46%	2.00	3.36	40.14%	1.90	2.75	39.10%

**Table 7 sensors-25-06698-t007:** Zero-shot prediction results of LLM-based methods.

Scenarios	OFA		GATGPT		GCNGPT		LLAMA2		ST-LLM		GSF-LLM	
	MAE	RMSE	MAE	RMSE	MAE	RMSE	MAE	RMSE	MAE	RMSE	MAE	RMSE
NYCTaxi Pick-up → NYCTaxi Drop-off	9.99	20.22	10.00	21.16	11.03	21.86	11.02	22.34	9.31	18.68	8.82	15.44
NYCTaxi Pick-up → CHBike Pick-up	3.61	5.98	3.29	5.60	3.53	5.91	3.25	5.15	3.06	5.40	2.83	4.72
NYCTaxi Pick-up → CHBike Drop-off	3.57	5.72	3.25	5.34	3.49	5.64	3.23	5.74	3.12	5.01	3.04	4.78
NYCTaxi Drop-off → NYCTaxi Pick-up	10.04	17.72	9.67	17.76	8.09	14.58	11.14	20.57	8.02	13.21	7.90	10.83
NYCTaxi Drop-off → CHBike Drop-off	3.58	5.72	3.19	4.99	3.35	5.19	3.29	4.99	3.09	4.65	2.75	4.51
NYCTaxi Drop-off → CHBike Pick-up	3.62	5.99	3.26	5.27	3.43	5.49	3.33	5.32	3.02	5.18	2.93	5.01

## Data Availability

The data used in this paper were collected from the NYCTaxi and CHBike online. Websites: https://www.nyc.gov/site/tlc/about/tlc-trip-record-data.page (accessed on 1 January 2025) and https://citibikenyc.com/system-data (accessed on 1 January 2025).

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
