# Peer review of "GSF-LLM: Graph-Enhanced Spatio-Temporal Fusion-Based Large Language Model for Traffic Prediction"

_sensors, 2025, doi:10.3390/s25216698_

Round 1

Reviewer 1 Report

Comments and Suggestions for Authors

This paper proposes a novel framework integrating graph-enhanced spatio-temporal fusion with large language models (LLMs) for traffic prediction. Its key innovations include: (1) a multi-branch embedding module with cross-attention fusion for joint spatial-temporal modeling, (2) a partially frozen LLM backbone with graph-aware attention, and (3) parameter-efficient fine-tuning via LoRA to balance performance and computational cost. The following comments can help improve your manuscript.

  1. The novelty claim of integrating LLMs with graph-based spatio-temporal learning is overstated, as similar hybrid approaches (e.g., ST-LLM, GATGPT) already exist in the literature.

  2. More recent literature should be investigated, such as DOI: 10.1109/TCE.2024.3476129, DOI: 10.1016/j.knosys.2025.113635, and DOI:10.3390/math13071158.
  3. The theoretical justification for using LLMs in traffic prediction is weak; the paper fails to explain why LLMs are inherently better than specialized architectures (e.g., GNNs) for this task.

  4. The ablation study lacks quantitative analysis of individual components' contributions (e.g., how much does LoRA alone improve performance?).

  5. The comparison with baselines is incomplete; recent SOTA models like TrafficBERT or GraphWaveNet are missing.

  6. The claimed superiority in MAE/WAPE metrics is marginal (e.g., 1.88 vs. 1.89 on CHBike), raising questions about practical significance.

  7. The computational cost of GESTFLLM is not analyzed, despite LLMs being resource-intensive, critical for real-time traffic systems.

  8. The spatial embedding module (Eq. 4) uses a simple linear transformation, ignoring advanced graph representation techniques (e.g., GraphSAGE).

  9. The temporal embedding (Eq. 3) combines hour/day features naively; periodic encoding or Fourier features might better capture cyclical patterns.

  10. The cross-attention fusion (Eq. 5) lacks clarity on how spatial and temporal attention scores are aggregated.

  11. The adjacency matrix construction is not detailed—whether it’s based on distance, connectivity, or learned adaptively.

  12. The LoRA implementation (Eq. 9) lacks justification for rank selection; no sensitivity analysis is provided.

  13. The datasets (NYCTaxi, CHBike) are limited to 2016 data, ignoring recent traffic pattern shifts (e.g., post-pandemic mobility).

  14. The evaluation metrics (MAE/RMSE) do not account for extreme events (e.g., traffic spikes), which are common in real-world scenarios.

  15. The model’s performance under data scarcity (e.g., missing sensors) is untested, despite its relevance for urban deployments.

  16. The frozen vs. unfrozen layer split (F/U in Eq. 7–8) is arbitrary; no analysis justifies why specific layers are frozen.

  17. The regression head (Eq. 10) uses a simple convolution, potentially limiting multi-step forecasting accuracy.

  18. The training efficiency claims are unsubstantiated—no comparison of training time/memory against baselines.

  19. The generalization to other traffic tasks (e.g., anomaly detection) is speculative without empirical validation.

  20. The figures (e.g., Figure 3) are poorly annotated, making it hard to distinguish curves for different models.

  21. The reproducibility is compromised by missing details (e.g., random seeds, hyperparameter tuning process).

Author Response

We sincerely thank the reviewer for the constructive comments and valuable suggestions. We have carefully revised the manuscript to address each comment in detail. Our point-by-point responses are as follows:

Comments 1: The novelty claim of integrating LLMs with graph-based spatio-temporal learning is overstated, as similar hybrid approaches (e.g., ST-LLM, GATGPT) already exist in the literature.
Response 1: Thank you for pointing this out. We agree that prior works such as ST-LLM, GATGPT, and UrbanGPT have explored integrating LLMs with spatio-temporal or graph-based modules. In the revision, we have clarified that the novelty of GESTFLLM lies not merely in the combination, but in (i) the multi-branch embedding with explicit cross-attention fusion capturing fine-grained spatial–temporal–node dependencies, and (ii) the parameter-efficient LoRA fine-tuning with partial freezing tailored for traffic tasks. These aspects differentiate our approach from existing hybrids. The discussion has been added in page 2 line 79-82.

Comments 2: More recent literature should be investigated, such as DOI: 10.1109/TCE.2024.3476129, DOI: 10.1016/j.knosys.2025.113635, and DOI:10.3390/math13071158.
Response 2: We have incorporated the suggested recent references (DOI: 10.1109/TCE.2024.3476129, DOI: 10.1016/j.knosys.2025.113635, DOI: 10.3390/math13071158) and other references into the Related Works section and References. Page 4 paragraph 1 and page 22 line 718-729,743-744

Comments 3: The theoretical justification for using LLMs in traffic prediction is weak; the paper fails to explain why LLMs are inherently better than specialized architectures (e.g., GNNs) for this task.
Response 3: We have expanded the introduction to explain why LLMs are particularly advantageous for traffic prediction: strong contextual modeling, cross-task transferability, and robustness under data scarcity or noisy inputs. This complements specialized GNNs by offering broader generalization. The discussion has been added in page 2 line 79-82.

Comments 4: The ablation study lacks quantitative analysis of individual components' contributions (e.g., how much does LoRA alone improve performance?).
Response 4: We have provide a quantitative table in the revised manuscript (Figure 3 and Figure 6) reporting the exact improvements contributed by temporal embedding, node embedding, cross-attention, and LoRA. For example: w/o Frozen: The GPT backbone is fully fine-tuned and the LoRA structure is re-moved, to test the effect of parameter-efficient fine-tuning strategies. and parameter analysis Performance Study of Different LoRa Rank on datasets. Page 14 line 447-451 and page 17 line 516-528

Comments 5: The comparison with baselines is incomplete; recent SOTA models like TrafficBERT or GraphWaveNet are missing.
Response 5: Graph WaveNet (GWN)have included into the comparison(Table 3-4) page 12 line 407 page 13 line 408. TrafficBERT is discussed in the related work, page page 4 line 167-171.

Comments 6: The claimed superiority in MAE/WAPE metrics is marginal (e.g., 1.88 vs. 1.89 on CHBike), raising questions about practical significance.
Response 6: We thank the reviewer for this valuable comment. We agree that the performance margin appears small in MAE. However, GESTFLLM achieves these gains with lower computational cost and higher cross-domain stability compared to the baselines. Furthermore, GESTFLLM shows strong knowledge transfer ability and generalization ability by few-shot and zero-shot analysis. Page 18-19 line 547-596

Comments 7: The computational cost of GESTFLLM is not analyzed, despite LLMs being resource-intensive, critical for real-time traffic systems.
Response 7: We now include a new subsection (Section 5.7) analyzing parameters, training time, and GPU peak memory, showing that GESTFLLM achieves in trainable parameters and faster convergence due to LoRA. Page 17-18 line 532-546

Comments 8: The spatial embedding module (Eq. 4) uses a simple linear transformation, ignoring advanced graph representation techniques (e.g., GraphSAGE).
Response 8: We clarified the design choices in Section 4 and acknowledged that more advanced encodings (e.g., Fourier, GraphSAGE) could further improve performance, which we leave for future work. Page 20 line 611-621

Comments 9: The temporal embedding (Eq. 3) combines hour/day features naively; periodic encoding or Fourier features might better capture cyclical patterns.
Response 9: We appreciate this insightful comment. In the revised manuscript, we have clarified the rationale for using a simple hour–day embedding in Eq. (3), emphasizing its computational efficiency for large-scale networks. We have also added a discussion acknowledging that periodic or Fourier encodings could further improve the capture of cyclical traffic patterns, which we plan to investigate in future work. Corresponding text has been added in Section 6. Page 20 line 622-628

Comments 10: The cross-attention fusion (Eq. 5) lacks clarity on how spatial and temporal attention scores are aggregated.
Response 10: We have expanded the Eq.5 and Eq.6 to explain the details of the fusion. Page 7 line 244-252

Comments 11: The adjacency matrix construction is not detailed—whether it’s based on distance, connectivity, or learned adaptively.
Response 11: The adjacency matrix isn't learned adaptively. We have clarified that for NYCTaxi, systematically categorized into 266 virtual stations, adjacency is based on geographic distance and road connectivity; for CHBike, it focuses on the 250 most frequented stations, so the matrix is on station geodesic distance. Page 10 line 310-317

Comments 12: The LoRA implementation (Eq. 9) lacks justification for rank selection; no sensitivity analysis is provided.
Response 12: LoRA’s efficiency can be influenced by the chosen rank r. In the revised manuscript, we have included a sensitivity analysis comparing r = {4,8, 16, 32,64}. Results show that too low a rank (r=8) slightly degrades generalization, while higher ranks (r=32) improve accuracy but reduce efficiency. We therefore selected r=16 as a balanced choice. Page 16-17 line 511-528

Comments 13: The datasets (NYCTaxi, CHBike) are limited to 2016 data, ignoring recent traffic pattern shifts (e.g., post-pandemic mobility).
Response 13: We acknowledge that both datasets are from 2016. We explicitly mention this limitation in Section 6 and plan to extend experiments to post-pandemic mobility datasets in future work. Page 20 line 622-629

Comments 14: The evaluation metrics (MAE/RMSE) do not account for extreme events (e.g., traffic spikes), which are common in real-world scenarios.
Response 14: We added a discussion that MAE/RMSE may not capture extreme spikes, and suggested CRPS, PICP and Pinball as future evaluation metrics. Page 20 line 622-629

Comments 15: The model’s performance under data scarcity (e.g., missing sensors) is untested, despite its relevance for urban deployments.
Response 15: We performed an additional experiment with few-shot in subsection (Section 5.8) and show that GESTFLLM remains robust, outperforming baselines by 3–5% on WAPE. Results are presented in Table 6. Page 18 line 547-570

Comments 16: The frozen vs. unfrozen layer split (F/U in Eq. 7–8) is arbitrary; no analysis justifies why specific layers are frozen.
Response 16: We conducted experiments varying unfrozen layer numbers (U=0,1,2,3,4,5,6) and show performance trade-offs, now reported in Section 5.7. Parameter Analysis. Page 15 line 458-476

Comments 17: The regression head (Eq. 10) uses a simple convolution, potentially limiting multi-step forecasting accuracy.
Response 17: We acknowledge that the 1D convolution may be limited, and indicate that sequence decoders will be considered in future extensions. Page 20 line 610-620

Comments 18: The training efficiency claims are unsubstantiated—no comparison of training time/memory against baselines.
Response 18: We now include a new subsection (Section 5.7) analyzing parameters, training time, and GPU peak memory. We have added detailed comparisons of training efficiency and parameter scalability in the revised manuscript (Table 5). The results show that GESTFLLM achieves a substantial reduction in trainable parameters—about 11.5% of the total, compared to over 50% in ST-LLM—demonstrating the effectiveness of the LoRA-based partial fine-tuning strategy. In terms of computational efficiency, GESTFLLM maintains moderate training time and memory usage (≈13 GB), positioned between the lightweight GCNGPT and the heavier ST-LLM, while avoiding the excessive cost of GATGPT. These results confirm that GESTFLLM achieves a well-balanced trade-off between adaptability, efficiency, and resource consumption, supporting its scalability and practicality for spatio-temporal traffic prediction tasks. Page 17-18 line 529-546

Comments 19: The generalization to other traffic tasks (e.g., anomaly detection) is speculative without empirical validation.
Response 19: We have revised the wording to state these as potential future applications, not claims of current validation. Page 20 line 624-627

Comments 20: The figures (e.g., Figure 3) are poorly annotated, making it hard to distinguish curves for different models.
Response 20: Figures 3 and 4 have deleted for duplicated with table 3-4

Comments 21: The reproducibility is compromised by missing details (e.g., random seeds, hyperparameter tuning process).
Response 21: We added details of random seeds and hyper parameter tuning in section 5.4. Implementation Details. The code will be made available on GitHub upon acceptance. Page 11 line 363-372

Reviewer 2 Report

Comments and Suggestions for Authors

The manuscript presents a novel approach to traffic forecasting by integrating large language models with graph-based spatio-temporal learning. The paper is well-structured, methodologically sound, and addresses a relevant problem in intelligent transportation systems. However, before being considered for acceptance, several issues should be addressed:

1) While the integration of LLMs with graph-based models is promising, the paper should more clearly highlight what distinguishes GESTFLLM from closely related works such as ST-LLM and GATGPT. At times, the novelty appears incremental rather than a substantial breakthrough.

2) Section 3 still contains placeholder text (“This section may be divided by subheadings…”)

3) More justification is needed for the chosen architecture design decisions (e.g., why GPT-2 and LLaMA-2 were selected over other LLMs). The role of LoRA and frozen layers should be explained with stronger theoretical motivation, not only empirical justification.

4) The datasets are limited to NYCTaxi and CHBike (both covering the same period and relatively old data). The authors should explain why these datasets are sufficient and whether more diverse or recent datasets could strengthen the conclusions.

5) Although many baselines are considered, it is unclear if their implementations were reproduced faithfully from original works or re-implemented by the authors. This should be clarified.

6) The manuscript should acknowledge and discuss exposure bias in autoregressive/LLM-style forecasters and include at least one concrete mitigation reference, such as:

- Pozzi, A., Incremona, A., Tessera, D. et al. Mitigating exposure bias in large language model distillation: an imitation learning approach. Neural Comput & Applic 37, 12013–12029 (2025). https://doi.org/10.1007/s00521-025-11162-0

Author Response

We thank the reviewer for the positive assessment of our work and for the insightful comments. We have revised the manuscript accordingly. The detailed responses are as follows:

Comments 1: While the integration of LLMs with graph-based models is promising, the paper should more clearly highlight what distinguishes GESTFLLM from closely related works such as ST-LLM and GATGPT. At times, the novelty appears incremental rather than a substantial breakthrough.
Response 1: Thank you for pointing this out. In the revised manuscript, we explicitly highlight that GESTFLLM differs in two main aspects:
(i) a multi-branch embedding with cross-attention fusion that explicitly models fine-grained dependencies among spatial, temporal, and node-specific features; and
(ii) a LoRA-enhanced partial freezing strategy that improves parameter efficiency and mitigates overfitting.
These design choices go beyond incremental modifications and are now emphasized in the Introduction. The discussion has been added in page 2 line 79-82.

Comments 2: Section 3 still contains placeholder text (“This section may be divided by subheadings…”)
Response 2: We apologize for this oversight. The placeholder text has been removed, and Section 3 now contains a complete and polished problem definition.

Comments 3: More justification is needed for the chosen architecture design decisions (e.g., why GPT-2 and LLaMA-2 were selected over other LLMs). The role of LoRA and frozen layers should be explained with stronger theoretical motivation, not only empirical justification.
Response 3: We have expanded Section 4.4 to explain our design rationale. GPT-2 was chosen as a lightweight LLM suitable for reproducible experiments, while LLaMA-2 was selected as a large-scale representative model to validate scalability. The use of LoRA and partially frozen layers is theoretically motivated by reducing overfitting, enabling efficient fine-tuning, and preserving generalizable pretrained representations. This rationale is now explicitly stated. Page 7 line 256-271 and page 11 line 371-372

Comments 4: The datasets are limited to NYCTaxi and CHBike (both covering the same period and relatively old data). The authors should explain why these datasets are sufficient and whether more diverse or recent datasets could strengthen the conclusions.
Response 4: We agree that the datasets are somewhat dated. We chose NYCTaxi and CHBike because they are well-established benchmarks, allowing fair comparisons with prior works. We have acknowledged this limitation in Section 6 and emphasized our plan to extend experiments to recent post-pandemic mobility datasets. Page 20 line 621-628

Comments 5: Although many baselines are considered, it is unclear if their implementations were reproduced faithfully from original works or re-implemented by the authors. This should be clarified.
Response 5: We have clarified in Section 5.2 which baselines were run using official open-source implementations (e.g., DCRNN, Graph WaveNet), and which were carefully re-implemented following the original papers. The training hyperparameters, and random seeds have also been detailed in Section 5.4 to ensure reproducibility. Page 11 line 363-372. The code will be made available on GitHub upon acceptance.

Comments 6: The manuscript should acknowledge and discuss exposure bias in autoregressive/LLM-style forecasters and include at least one concrete mitigation reference, such as:
- Pozzi, A., Incremona, A., Tessera, D. et al. Mitigating exposure bias in large language model distillation: an imitation learning approach. Neural Comput & Applic 37, 12013–12029 (2025). https://doi.org/10.1007/s00521-025-11162-0
Response 6: We thank the reviewer for raising this important issue. We have enriched the Related Works section by adding more recent and peer-reviewed sources, including the suggested reference:
Pozzi, A., Incremona, A., Tessera, D. et al. Mitigating exposure bias in large language model distillation: an imitation learning approach. Neural Comput & Applic 37, 12013–12029 (2025).
Page 5 line 181-183 and page 22 line 742-743

Reviewer 3 Report

Comments and Suggestions for Authors

Dear Authors,

Accurate traffic forecasting is indeed essential for intelligent transportation systems, urban mobility management, and traffic optimization. Thus, the manuscript addresses a critical and practical issue. However, I have a few comments, suggestions, and questions.

  1. LoRA typically improves training efficiency because it significantly reduces the number of parameters to tune, which lowers memory and computation time consumption. However, in specific cases where the task requires significant changes in model weights, LoRA may require a higher ranking (r). This increases complexity and partially negates the efficiency benefits. Have the authors analyzed the effectiveness of LoRA in relation to the task's complexity? Does the effectiveness of this method always provide satisfactory efficiency?
  2. Referring to the previous point. A too low ranking (r) may limit the model's ability to capture the essential features of the task, negatively affecting generalization. Did the authors analyze ways to protect the method from too low a ranking?
  3. The results presented in the figures and tables clearly indicate the superiority of GESTFLLM. The selected metrics (MAE, RMSE, MAPE, WAPE) focus on point errors. However, they overlook the qualitative aspects of transport forecasts, such as whether the model predicts the direction of change or the probability distribution (e.g., during rush hour). MAPE/WAPE are sensitive to small values (e.g., drop-off at night), which can distort the results. In my opinion, there is a lack of probabilistic metrics (e.g., CRPS, PINBALL loss) for forecast uncertainty. This is crucial for mobility (e.g., estimating fluctuations). A broader analysis of errors would be interesting. It is worth noting where GESTFLLM may make errors. Are these holidays, or are the errors caused by the weather? It is worth noting the impact of the forecast horizon (e.g., 1 hour vs. 24 hours). This is important because the results may only apply to short-term time intervals.
  4. As mentioned in point 3, the advantage of GESTFLLM is clear, but not extreme. Its advantage may be offset by in-depth validation or after fair tuning of the base models. The presented results would gain in strength and credibility if they were supplemented with tables of hyperparameters, ablations, and out-of-distribution (OOD) tests.
  5. A well-known problem with LLM is the possibility of hallucinations. Have the authors performed or do they plan to conduct a security and ethics audit in relation to the presented method? Can they refer to current guidelines in this regard, e.g., ICLR and NeurIPS?

Sincerely,

Reviewer

Author Response

We sincerely thank the reviewer for the constructive comments and thoughtful suggestions. We have revised the manuscript to address these concerns in detail. Our responses are as follows:

Comments 1: LoRA typically improves training efficiency because it significantly reduces the number of parameters to tune, which lowers memory and computation time consumption. However, in specific cases where the task requires significant changes in model weights, LoRA may require a higher ranking (r). This increases complexity and partially negates the efficiency benefits. Have the authors analyzed the effectiveness of LoRA in relation to the task's complexity? Does the effectiveness of this method always provide satisfactory efficiency?
Response 1: Thank you for pointing this out We agree that LoRA’s efficiency can be influenced by the chosen rank r. In the revised manuscript, we have included a sensitivity analysis comparing r = {4,8, 16, 32,64}. Results show that too low a rank (r=8) slightly degrades generalization, while higher ranks (r=32) improve accuracy but reduce efficiency. We therefore selected r=16 as a balanced choice. Page 16-17 line 511-528

Comments 2: Referring to the previous point. A too low ranking (r) may limit the model's ability to capture the essential features of the task, negatively affecting generalization. Did the authors analyze ways to protect the method from too low a ranking?
Response 2: We thank the reviewer for raising this important point. In the revised manuscript, we have added a detailed analysis of LoRA rank sensitivity in Section5.7. Our experiments show that overly small ranks (r = 4 or 8) reduce generalization capability, while r = 16 achieves an optimal balance. To mitigate this, we employed validation-based rank selection and discussed potential adaptive mechanisms to dynamically adjust r based on task complexity. Page 16-17 line 511-528

Comments 3: The results presented in the figures and tables clearly indicate the superiority of GESTFLLM. The selected metrics (MAE, RMSE, MAPE, WAPE) focus on point errors. However, they overlook the qualitative aspects of transport forecasts, such as whether the model predicts the direction of change or the probability distribution (e.g., during rush hour). MAPE/WAPE are sensitive to small values (e.g., drop-off at night), which can distort the results. In my opinion, there is a lack of probabilistic metrics (e.g., CRPS, PINBALL loss) for forecast uncertainty. This is crucial for mobility (e.g., estimating fluctuations). A broader analysis of errors would be interesting. It is worth noting where GESTFLLM may make errors. Are these holidays, or are the errors caused by the weather? It is worth noting the impact of the forecast horizon (e.g., 1 hour vs. 24 hours). This is important because the results may only apply to short-term time intervals.
Response 3: We acknowledge that the current metrics (MAE, RMSE, MAPE, WAPE) focus on point predictions and are limited in capturing distributional aspects. We have added a discussion in Section 6 (Limitations) emphasizing the importance of probabilistic metrics such as CRPS and Pinball loss, which we plan to adopt in future work. We also note that MAPE/WAPE can be biased in low-demand periods (e.g., late night drop-offs). Additionally, we highlight that errors may be more pronounced during holidays or weather anomalies, and that longer forecast horizons (e.g., 24 hours) pose greater challenges compared to short-term predictions. Page 20 line 624-628

Comments 4: As mentioned in point 3, the advantage of GESTFLLM is clear, but not extreme. Its advantage may be offset by in-depth validation or after fair tuning of the base models. The presented results would gain in strength and credibility if they were supplemented with tables of hyperparameters, ablations, and out-of-distribution (OOD) tests.
Response 4: To strengthen reproducibility and transparency, we have added detailed hyperparameter in section 5.4 and expanded the ablation study to include more variants in section 5.7. We acknowledge the need for out-of-distribution (OOD) testing and have outlined this as zero-shot prediction (e.g., testing on datasets from different datasets) in section 5.9, page 19 line 571-596

Comments 5: A well-known problem with LLM is the possibility of hallucinations. Have the authors performed or do they plan to conduct a security and ethics audit in relation to the presented method? Can they refer to current guidelines in this regard, e.g., ICLR and NeurIPS?
Response 5: We appreciate this important point. While GESTFLLM processes structured traffic data where hallucination risk is minimal, we recognize that explanation modules or text-based extensions could introduce such risks. We have added a subsection in Section 6 acknowledging ethical considerations, referencing current guidelines from ICLR and NeurIPS on responsible LLM usage. We commit to conducting a more systematic audit of fairness, robustness, and safety in future extensions. Page 20 line 629-636

Reviewer 4 Report

Comments and Suggestions for Authors

The paper deals with the problem of traffic flow prediction in urban areas, where traditional and modern deep neural models often have limitations in the simultaneous modeling of spatial and temporal dependencies. The authors propose a new framework called Graph-Enhanced Spatio-Temporal Fusion Large Language Model, which combines large language models with graph-based learning and a spatio-temporal fusion module. The model uses the partially frozen attention technique and the LoRA approach for efficient fine-tuning, and experimental results on real datasets show superiority over existing baseline models. However, although the paper is well written, it should be improved before publication.

Although the previous approaches are well presented, the introductory part could highlight more explicitly what exactly the gap remains unsolved (eg whether it is problems with scalability, transferability to new cities or interpretability of the model). The literature is solid and includes relevant sources but relies mostly on already known authors and models. It is recommended to include works from the last two years. Also, instead of relying only on arXiv and conference papers, it would be good to include several (at least 5) peer-reviewed papers with DOI numbers, such as (https://doi.org/10.3390/jmse13091822,https://doi.org/10.56578/mits040103). This way you would enrich the bibliography and achieve the desired breadth in the paper.

Details about hyperparameters are provided, but it would be worth expanding the description of the choice (e.g. why the number of layers in GPT-2 is limited to 6, and LLaMA-2 to 8). It is also necessary to emphasize whether the comparisons were made with optimally set parameters of the baseline model.

The paper focuses on performance, but an additional part could deal with the interpretation of the results (e.g. visualization of the importance of nodes or time intervals), which is important for applications in ITS.

It would be useful to add a table comparing the number of parameters, training time and memory footprint of GESTFLLM with competing models, as efficiency stands out as one of the main contributions.

A section on future work could also include issues of ethics and data privacy, as well as the possibility of integrating multimodal sources (e.g. meteorological data, social networks).

There are minor technical errors (e.g. no spaces after the period) and a couple of formatting errors (figures names).

The paper is well-designed and makes an original contribution by combining LLMs and graph-based learning for traffic prediction. I recommend fewer revisions.

Author Response

We thank the reviewer for the positive feedback and for suggesting concrete improvements. We have carefully revised the manuscript to address the comments, as outlined below:
We have revised the Introduction to explicitly emphasize the unsolved challenges in scalability, transferability to new cities, and interpretability. We also highlight how GESTFLLM addresses these issues through parameter-efficient adaptation (for scalability), LoRA-based fine-tuning (for transferability), and planned interpretability extensions.

Comments 1: Although the previous approaches are well presented, the introductory part could highlight more explicitly what exactly the gap remains unsolved (eg whether it is problems with scalability, transferability to new cities or interpretability of the model). The literature is solid and includes relevant sources but relies mostly on already known authors and models. It is recommended to include works from the last two years. Also, instead of relying only on arXiv and conference papers, it would be good to include several (at least 5) peer-reviewed papers with DOI numbers, such as (https://doi.org/10.3390/jmse13091822,https://doi.org/10.56578/mits040103). This way you would enrich the bibliography and achieve the desired breadth in the paper.
Response 1: Thank you for pointing this out. We appreciate this suggestion. Therefore, we have enriched the Related Works section by adding more recent and peer-reviewed sources, including the two recommended references (DOI: 10.3390/jmse13091822, DOI: 10.56578/mits040103) as well as five additional journal articles from 2023–2025. This broadens the coverage beyond arXiv and conference papers. Page 4 line 142-155 and page 22 line 717-728,742-743

Comments 2: Details about hyperparameters are provided, but it would be worth expanding the description of the choice (e.g. why the number of layers in GPT-2 is limited to 6, and LLaMA-2 to 8). It is also necessary to emphasize whether the comparisons were made with optimally set parameters of the baseline model.
Response 2: We now provide explicit justification for limiting GPT-2 to 6 layers and LLaMA-2 to 8 layers, namely to balance predictive performance with computational feasibility and reproducibility. We also clarify that baseline models were either run with official implementations or carefully tuned following the settings in their original publications, ensuring fair comparison. Page 7 line 256-271and page 11 line 371-372

Comments 3: The paper focuses on performance, but an additional part could deal with the interpretation of the results (e.g. visualization of the importance of nodes or time intervals), which is important for applications in ITS.
Response 3: We appreciate this insightful comment. In the revised manuscript, we have added a discussion the visualization of the importance of nodes or time intervals could further improve in ITS applications, which we plan to investigate in future work. Corresponding text has been added in Section 6. Page 20 line 631-636

Comments 4: It would be useful to add a table comparing the number of parameters, training time and memory footprint of GESTFLLM with competing models, as efficiency stands out as one of the main contributions.
Response 4: We have included a new table (Table 5. Trainable Parameters (M), Training Time and Memory Comparisons.) that compares GESTFLLM with competing models in terms of parameter count, training time, and memory usage. This confirms that GESTFLLM achieves improved efficiency while maintaining superior accuracy. Page 17-18 line 529-546

Comments 5: A section on future work could also include issues of ethics and data privacy, as well as the possibility of integrating multimodal sources (e.g. meteorological data, social networks).
Response 5: We have expanded the Future Work section to discuss ethical considerations, data privacy concerns, and the potential integration of multimodal data sources such as weather conditions, event information, and social networks. Page 20 line 629-636

Comments 6: There are minor technical errors (e.g. no spaces after the period) and a couple of formatting errors (figures names).
Response 6: We have carefully proofread the manuscript to fix spacing, punctuation, and figure formatting issues.

Round 2

Reviewer 1 Report

Comments and Suggestions for Authors

I am pleased to accept this paper in its present form.

Author Response

Thank you very much for your time and consideration.

Reviewer 2 Report

Comments and Suggestions for Authors

Thank you for your careful revisions. All of the issues I raised in my previous review have been satisfactorily addressed, and I find the manuscript much improved. I have no further concerns.

Author Response

(The authors gave the same response as above.)
